# Arbitrarily Oriented Phase Randomization of Design Ground Motions by Continuous Wavelets

**Haoyu Xie** [1,2,*] **and Riki Honda** [3]

1   State Key Laboratory of Bridge Engineering Structural Dynamics,
    CMCT Research & Design Institute Co., Ltd., Chongqing 400067, China
2   School of Civil Engineering, Chongqing University, Chongqing 400045, China
3   Graduate School of Frontier Science, The University of Tokyo, Kashiwa 277-8561, Japan;
    rikihonda@k.u-tokyo.ac.jp
*   Correspondence: xhy.civil@yahoo.com

**Abstract:** For dynamic analysis in seismic design, selection of input ground motions is of huge importance. In the presented scheme, complex Continuous Wavelet Transform (CWT) is utilized to simulate stochastic ground motions from historical records of earthquakes with phase disturbance arbitrarily localized in time-frequency domain. The complex arguments of wavelet coefficients are determined as phase spectrum and an innovative formulation is constructed to improve computational efficiency of inverse wavelet transform with a pair of random complex arguments introduced and make more candidate wavelets available in the article. The proposed methodology is evaluated by numerical simulations on a two-degree-of-freedom system including spectral analysis and dynamic analysis with Shannon wavelet basis and Gabor wavelet basis. The result shows that the presented scheme enables time-frequency range of disturbance in time-frequency domain arbitrarily oriented and complex Shannon wavelet basis is verified as the optimal candidate mother wavelet for the procedure in case of frequency information maintenance with phase perturbation.

**Keywords:** structural seismic design; dynamic analysis; input ground motion; wavelet transform; uncertainty





## 1. Introduction

In performance-based earthquake engineering, dynamic analysis is often utilized to evaluate seismic performance of target structures. As a slight fluctuation of input ground motion in time-history analysis results in huge difference of nonlinear structural response [1], uncertainty of design input ground motions should be considered, which raises a significant challenge [2,3].

Conventionally, input ground motions for seismic design are accessible from history records of previous earthquakes and artificial ground motion simulation technique by empirical relationships on fault models [4,5]. Numerical and empirical simulations generate design ground motions based on considered fault parameters, although specific parameters required for prediction cannot be determined accurately [6]. Meanwhile, due to the complexity of physical models for earthquake phenomenon, there is not a perfect empirical relationship to reproduce the exact ground motion.

Indices (intensity measures) based procedures assume that a ground motion which is large enough in terms of indices is supposed to be 'tough' for structures. Peak ground motion acceleration (PGA) is commonly used as an index to quantify the effectiveness of ground motions from past seismic records, however, considering the fact that the number of previous earthquake records by simple amplification cannot meet the request of diversity for input ground motions in seismic design, and it is hardly possible that the same earthquake could just happen twice at the particular target site, using ground motion from historical records as design ground motion is restrained. Response spectrum is another

index utilized for evaluation of input ground motion [7–10]. In current seismic design codes, acceleration response spectra-compatible ground motion is required for dynamic analysis, although it is not entirely suitable for situations for which nonlinear performance of the structures is dominant because dynamic behavior of nonlinear structures is more sensitive and complicated than it is described by response spectra in frequency domain.

As intensity measures cannot fully evaluate the complexity of nonlinear structural performance, another methodology is proposed to fluctuate phase spectrum of an original ground motion and generate artificial ground motions by Fourier Transform (FT), which is called stochastic method [11]. Discrete Wavelet Transform (DWT) is also adopted for time-frequency analysis in similar pattern for the purpose of synthesis of stochastic ground motions [12,13].

The objective of the present paper is to introduce a novel scheme to have time-frequency characteristics of an original signal fluctuated with arbitrary orientation of wavelet phase by using complex continuous wavelet transform and to compare candidates of mother wavelets for optimization of the scheme. Such a design input motion generated for dynamic analysis for seismic design could be considered highly stochastic and uncertainty compensated. With the higher diversity and the more information within input ground motions group in dynamic analysis, it is believed a better infrastructure could be built for seismic engineering.

## 2. Stochastic Method

Stochastic method utilizes FT to fluctuate the original ground motion's Fourier amplitude spectrum with a random phase spectrum disturbance. Modified ground motion retains some of the parametric and functional descriptions in frequency domain, which is still related to the earthquake magnitude and to the distance from the source.

### 2.1. Conventional Fourier Analysis

The scheme uses sinusoidal function $e^{i\omega t}$, basis of Fourier transform, that has shift-invariance and orthogonal properties. The first step of the procedure is to decompose a signal into amplitude spectrum and phase spectrum in frequency domain by discrete Fourier transform, which is given as:

$$F(f_k) = \sum_{n=0}^{N-1} signal(t_n)e^{-if_k t_n}\Delta t = |F(f_k)|e^{-i\theta(f_k)} \tag{1}$$

where $t_n$ denotes sampling points of the signal in time domain with sampling duration $\Delta t$ while $f_k$ denotes sampling points in frequency domain, and $|F(f_k)|$ denotes amplitude spectrum with $\theta(f_k)$ as phase spectrum of the original signal. After phase spectrum altered, inverse Fourier transform is conducted as:

$$\theta'(f_k) = \theta(f_k) + \sigma(f_k) \tag{2}$$

$$signal'(t_n) = \sum_{k=0}^{K-1} |F(f_k)|e^{-i\theta'(f_k)}e^{if_k t_n}\Delta f \tag{3}$$

where $\sigma(f_k)$ denotes artificial phase spectrum aiming at fluctuating the original phase spectrum $\theta(f_k)$. Stochastic method based on discrete Fourier transform has been considered as the most common methodology to generate artificial earthquakes with random phase spectrum.

Nevertheless, there is a major demerit of this methodology that the phase disturbance cannot be localized in time domain as Fourier coefficients could only contain the information of the frequency from the original signal.

*2.2. Modification Using DWT*

Wavelet transform aims to decompose a signal into a set of basis functions consisting of contractions, expansions, and translations of a mother function $\Psi(\tau, s)$ [14]. Due to its characteristics of revealing both time and frequency information, wavelet transform has been utilized recently as an alternative to Fourier transform in the process of artificial ground motion simulation. In usual cases, the concept of phase is not rigorously determined [15]. In the scheme, complex arguments of wavelet coefficients are regarded as phase spectrum, and decomposition of the signal by discrete wavelet transform is given as:

$$W(j,k) = \sum_{n=0}^{N-1} signal(t_n) \frac{1}{\sqrt{2^j}} \Psi\left(\frac{t_n - k2^j}{2^j}\right) \Delta t = |W(j,k)| e^{i\theta(j,k)} \tag{4}$$

where *j* represents scale (frequency domain) and *k* represents transition (time domain), as scale samples of wavelet transforms following a geometric sequence of ratio 2 in dyadic wavelet analysis. Sharing similar manner to Equation (2), phase spectrum in wavelet analysis is altered as:

$$\theta'(j,k) = \theta(j,k) + \sigma(j,k) \tag{5}$$

where $\sigma(j,k)$ denotes disturbance phase spectrum. The artificial signal is reconstructed by inverse wavelet transform as:

$$C_\Psi = \sum_{n=0}^{N-1} \frac{\left|\hat{\Psi}(\omega_n)\right|^2}{|\omega_n|} \Delta\omega \tag{6}$$

$$signal'(t_n) = C_\Psi^{-1} \Delta \sum_j \sum_k |W(j,k)| e^{i\theta'(j,k)} \frac{1}{\sqrt{2^j}} \Psi\left(\frac{t_n - k2^j}{2^j}\right) \Delta\left(k2^j\right) \frac{\Delta\left(2^j\right)}{4^j} \tag{7}$$

where $\hat{\Psi}(\omega_n)$ denotes Fourier transform of wavelet basis $\Psi(t_n)$, and $C_\Psi$ denotes the operator for inverse process of continuous wavelet transform.

Analytical discrete wavelet transform enables localized disturbances at desired time intervals, by maintaining shift-invariance and orthogonality as Stochastic Method and allows conducting ground motion simulation considering uncertainties in wavelet phase, although time-frequency localization is not arbitrarily oriented due to the characteristics of dyadic distribution of both transition and scale, and the uncertainty principle of time-frequency resolution.

## 3. Modified Inverse CWT

Continuous wavelet transform manages to randomize phase spectrum at any desired area in time-frequency domain, which cannot be realized by discrete wavelet transform as described in Section 2.2. Inverse wavelet transform is well defined by a double integral or sum in transition and scale domain as Equation (7). However, as processing of the method for continuous wavelets requires extremely high performance of the computer and the result precision relies on the certain distribution of sampling scales and shifts, application of the method is often beyond engineers' capability. Moreover, orthogonality property of wavelet basis is prescribed in conventionally defined inverse transform, so that some wavelet bases with some useful properties like highly compressed product of standard deviations but without orthogonality are unavailable in this research. Considering such problems, we proposed a scheme, by which inverse wavelet transform could be efficiently conducted after phase randomly altering. First, the signal is decomposed continuously in a desired time-frequency domain, which is represented as:

$$W(s, \tau) = \int signal(t) \frac{1}{\sqrt{s}} \Psi \left( \frac{t - \tau}{s} \right) \Delta t \tag{8}$$

where $s$ represents scale (frequency domain) and $\tau$ represents transition (time domain). Then, the original wavelet coefficients in the certain domain are replaced with coefficients fluctuated by random phase spectrum $\theta(s, \tau)$ based on Mersenne Twister. Assuming disturbance as the difference between two sets of wavelet coefficients, it is given as:

$$dis'_{s,\tau}(t_n) = -W(s, \tau) \frac{1}{\sqrt{s}} \Psi \left( \frac{t_n - \tau}{s} \right) c + W(s, \tau) e^{i\theta(s,\tau)} \frac{1}{\sqrt{s}} \Psi \left( \frac{t_n - \tau}{s} \right) c \tag{9}$$

where $dis'_{s,\tau}(t_n)$ denotes a single fluctuation on the original signal corresponding to the certain scale and transition $(s, \tau)$, and $c$ represents a constant coefficient influencing how much this disturbance is amplified. Requirement of computational high performance is avoided by the equation. Equation (9) is derived from:

$$dis'_{s,\tau}(t_n) = signal(t_n) - signal'(t_n) \tag{10}$$

considering for the fluctuation at the certain pair of scale and transition $(s, \tau)$ in time-frequency domain, from Equation (10) and Equation (7) by decomposing both $signal(t_n)$ and $signal'(t_n)$ into the sum of wavelet coefficients, we get Equation (9).

Since the diversity of total power distribution of the generated signal needs to be enhanced for higher randomness, an extra complex argument is introduced as a modification to Equation (9), and the artificial ground motion generated from the original signal is given as:

$$signal'(t_n) = signal(t_n) - \sum_s \sum_\tau W(s, \tau) \frac{1}{\sqrt{s}} \Psi \left( \frac{t_n - \tau}{s} \right) \left[ e^{i\theta_1(s,\tau)} - e^{i\theta_2(s,\tau)} \right] c \tag{11}$$

where $\theta_1(s, \tau)$ and $\theta_2(s, \tau)$ represent two random phase spectra (sets of complex arguments). Equation (11) could be understood as a process that phase spectrum of the original signal is reduced by phase spectrum $\theta_1(s, \tau)$ and then increased with phase spectrum $\theta_2(s, \tau)$. The procedure is not linear, so that the pair of random phase spectra in Equation (11) cannot be replaced by single phase spectrum.

Shannon wavelet and Gabor wavelet are selected as candidates of mother wavelet in the article due to appropriate qualities for this research. Complex Shannon wavelet is an analytic hardy wavelet with sinc function as its real and imaginary part which has no negative frequency component, so that total power of the signal is maintained in the scheme based on complex Shannon wavelet. Its real and imaginary parts are shown in Figure 1a and the same parts in frequency domain is shown in Figure 1b. Gabor wavelet minimizes the product of its standard deviations in the time and frequency domain, meaning that the uncertainty in information carried by this wavelet is minimized, although total power of the signal may not be maintained in the scheme based on Gabor wavelet. Real and imaginary parts of Gabor wavelet is shown in Figure 2a and the same parts in frequency domain is shown in Figure 2b.

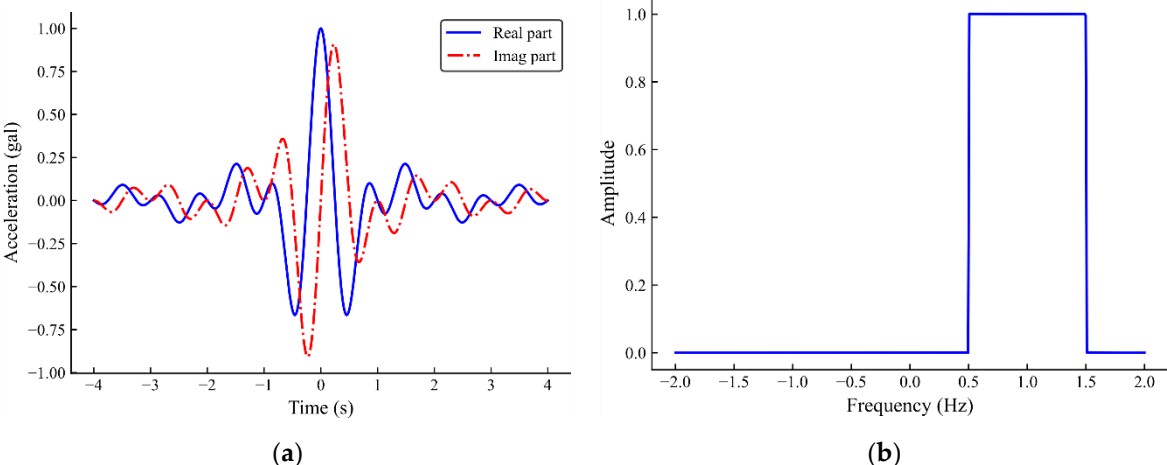

**Figure 1.** (**a**) Time and (**b**) frequency characteristics of complex Shannon wavelet basis.

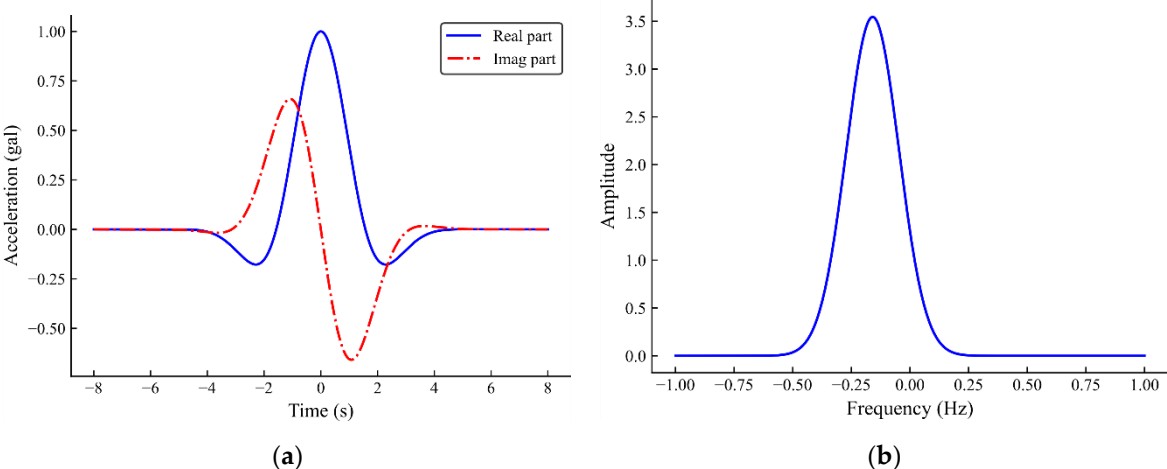

**Figure 2.** (**a**) Time and (**b**) frequency characteristics of Gabor wavelet basis.

## 4. Numerical Simulation

### 4.1. Spectral Analysis

The EW component of a strong ground motion data of 2000 Tottori Earthquake ($M_w$ 6.7) observed from KYT001 station of online database Kiban-Kyoshin-net [16] is selected for numerical simulation in the article as it is considered a typical urban earthquake. Amplitude spectra of the time series by Shannon wavelet transform and Gabor wavelet transform are shown in Figure 3.

In case of Figure 3, it is verified that both amplitude spectra share similar dominant range for contributing the most of total power of the signal in time-frequency domain (the scale is amplified according to the center frequency of mother wavelets, so that the scale in Shannon wavelet amplitude spectrum is four times the scale in Gabor wavelet amplitude spectrum). Based on the distribution of dominant range of wavelet amplitude spectra, in the article, 15 to 45 s transition and 0 to 3 scale in Shannon wavelet spectrum are selected as the target domain, while 15 to 45 s transition and 0 to 0.75 scale in Gabor wavelet spectrum are selected as the target domain. 10,000 wavelet coefficients in the target domain are chosen randomly to be disturbed by uncertainties of two sets of phase spectra $\theta_1(s, \tau)$ and $\theta_2(s, \tau)$ from 0 to $\pi$ in Equation (10). The disturbances generated by two wavelets based on Equation (9) with the certain set of phase spectra are shown in Figure 4, and response spectra of two artificial ground motions compared to original signal are shown in Figure 5.

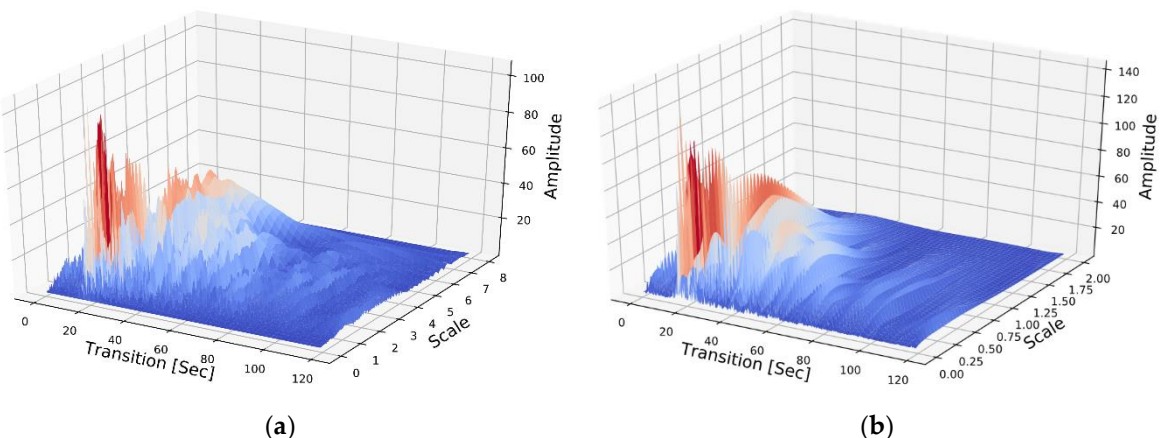

**Figure 3.** Amplitude spectra of the time series by (**a**) Shannon wavelets and (**b**) Gabor wavelets.

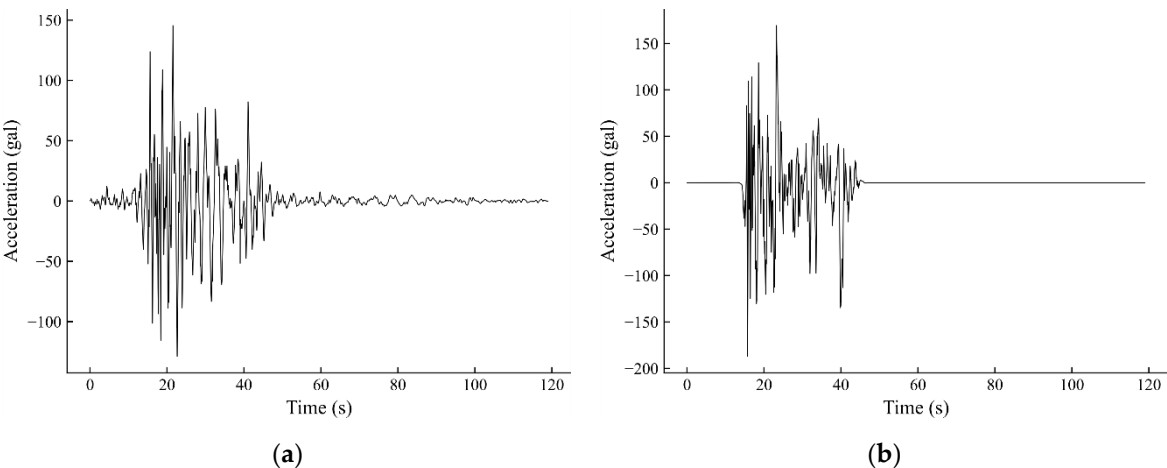

**Figure 4.** Random disturbance generated by (**a**) Shannon wavelets and (**b**) Gabor wavelets.

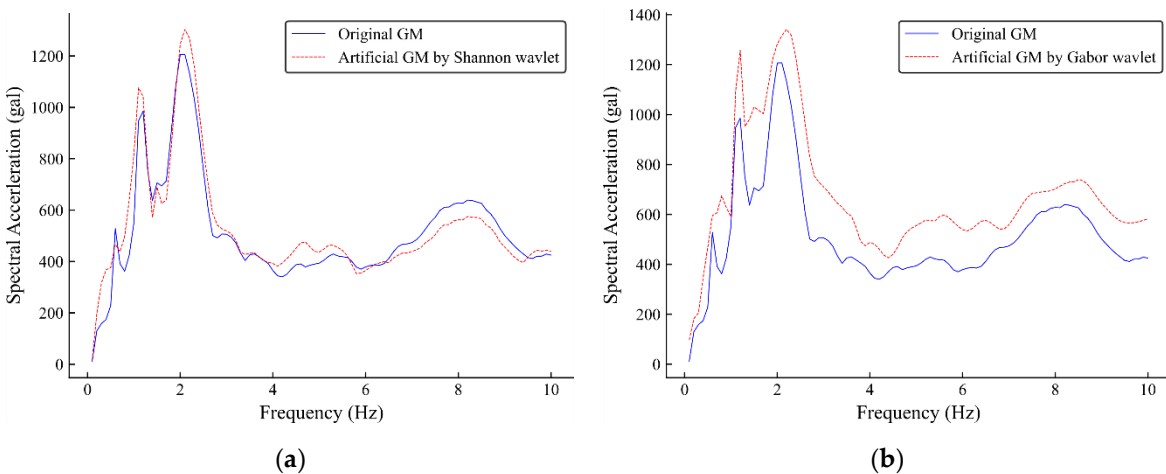

**Figure 5.** Comparison between response spectra of original ground motion and artificial ground motion generated by (**a**) Shannon wavelets and (**b**) Gabor wavelets.

It can be seen from Figure 4 that the disturbances generated by both wavelet bases are restrained in the certain time domain (15 to 45 s) as we designed. In contrast, Gabor wavelet provides higher performance of filtering noise in the domain aside from dominant time range due to the property of minimization the product of its standard deviations.

However, in case of Figure 5, it can be seen that only complex Shannon wavelet out of two candidates relatively manages to maintain the shape of response spectrum and power distribution of the ground motion in frequency domain, since it is an analytic wavelet. As it is in stochastic method, the amplitude information in frequency domain is considerably preserved by Shannon wavelet, while modification in the time domain of acceleration series and phase information in frequency domain leads to the diversity enhancement of the seismic responses to the certain ground motion set.

To distinctly illustrate the improvement by the proposed scheme compared to stochastic method in Section 2, the certain disturbance and artificial ground motion time series modified by stochastic method are shown in Figure 6. Contrast to Figure 4, the disturbance in Figure 6a by stochastic method is not restrained in the dominant time domain. The fluctuation in whole time domain cannot be averted by traditional stochastic method.

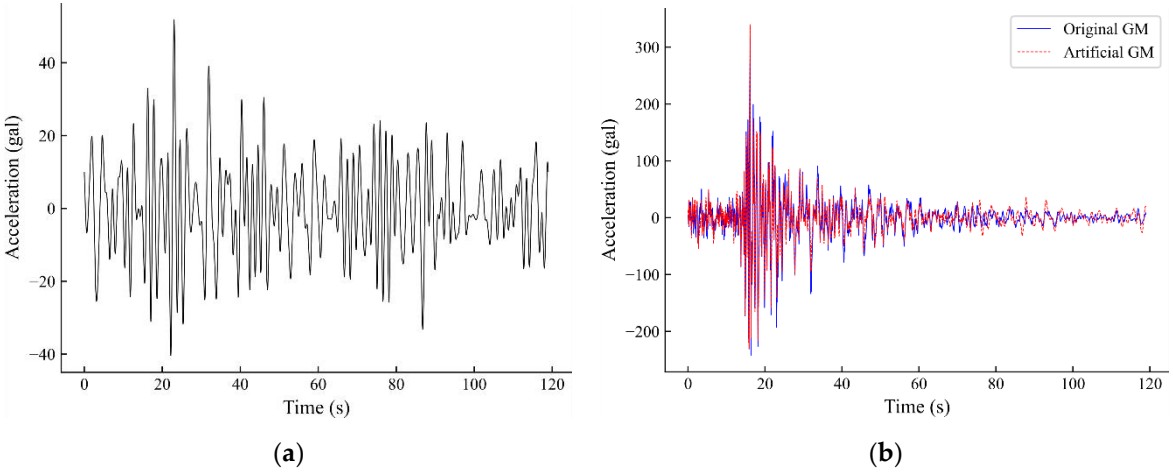

**Figure 6.** A case of artificial ground motion simulation by Stochastic Method: (**a**) Random disturbance; (**b**) Original and artificial time series.

### 4.2. Dynamic Analysis

Based on the above, complex Shannon wavelet is considered the optimal mother wavelet out of two candidates due to its power maintenance with phase perturbation. Dynamic analysis is to be conducted with complex Shannon wavelet in this section.

An idealized 2-story shear-frame-model is used here as the target structure. The building is simulated by a 2-degree-of-freedom model in which nonlinear behavior of springs is expressed by tri-linear Clough model, modified following a case in Architectural Institute of Japan [17] as shown in Figure 7. The seismic response of the structure is simulated in software Open System for Earthquake Engineering Simulation [18] by utilizing time domain Newmark-$\beta$ method.

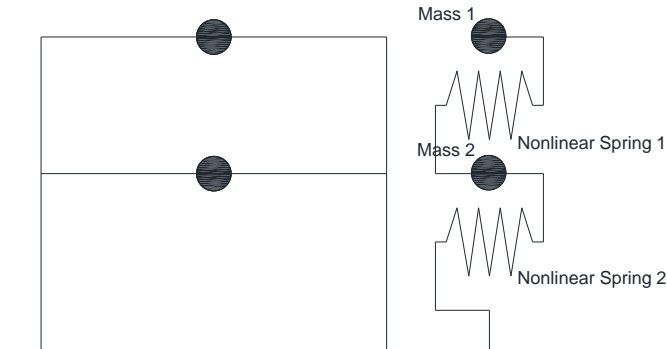

**Figure 7.** Target structure and analysis model corresponded.

Through modal analysis (initial stiffness is considered), modal frequencies and periods are obtained as shown in Table 1, according to which, 15 to 45 s and 0.4 to 0.6 Hertz in Shannon wavelet spectrum are selected as the target domain to disturb the certain phase spectrum of 1000 random wavelet coefficients. In a particular case, the acceleration time series of original signal and artificial signal are shown in Figure 8, and the vertical seismic deformation responses of the target structure are shown in Figure 9.

**Table 1.** Modal frequencies and periods of the analysis model.

|  | Frequency (Hz) | Period (s) |
|---|---|---|
| 1st Mode | 0.39 | 2.57 |
| 2nd Mode | 0.78 | 1.29 |

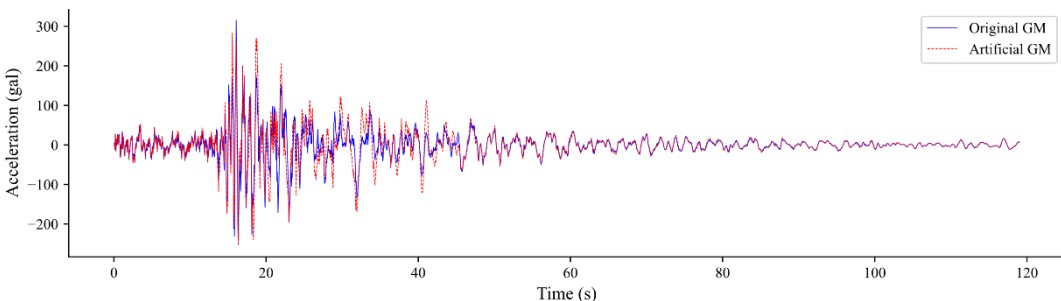

**Figure 8.** Acceleration time series of original signal and artificial signal.

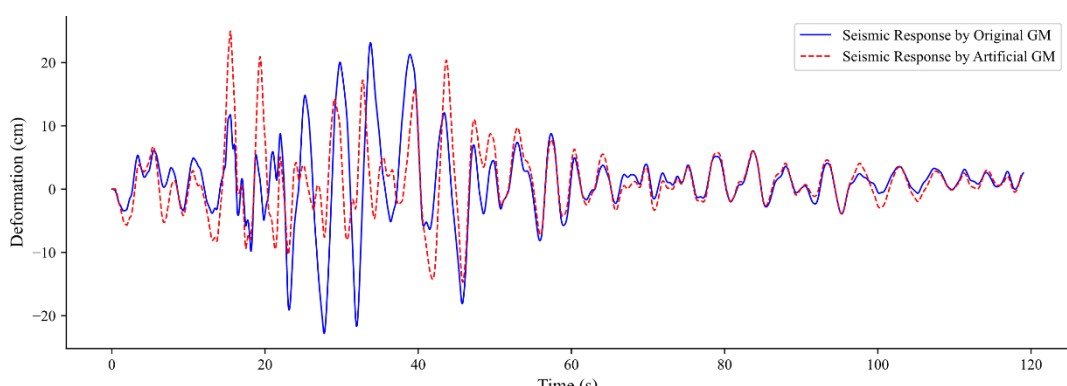

**Figure 9.** Vertical seismic responses of deformation of the target structure.

In case of Figures 8 and 9, it could be found that artificial ground motion generated by the proposed methodology excites completely different deformative response of the target structure, due to the random phase spectrum in the structural dominant time-frequency range.

Besides the certain case above, 50 sets of random phase spectra are generated to disturb the original signal by continuous wavelet transform. The same process is carried out by discrete wavelet transform as the baseline. Since CWT arbitrarily orients phase disturbance in frequency domain, the frequencies selected with CWT could be more corelated to modal frequencies of the target structure, and thus enables the methodology to be more efficient because artificial ground motion synthesized by the scheme is supposed to excite more variant seismic response of the target structure due to resonance. Maximum displacement of the target structure is selected as the intensity measure to evaluate seismic performance. The result shows that the mean square error of maximum displacements by DWT scheme is 1.97, while the mean square error of maximum displacements by CWT scheme is only 0.489. Figure 10 shows the distribution of two sets of maximum displacements responded to artificial ground motions representatively disturbed by CWT and DWT. As CWT explicitly amplifies the variance of seismic responses, it is considered that the diversity of input

ground motions is enhanced and the uncertainty in seismic design by dynamic analysis is decreased with the proposed method.

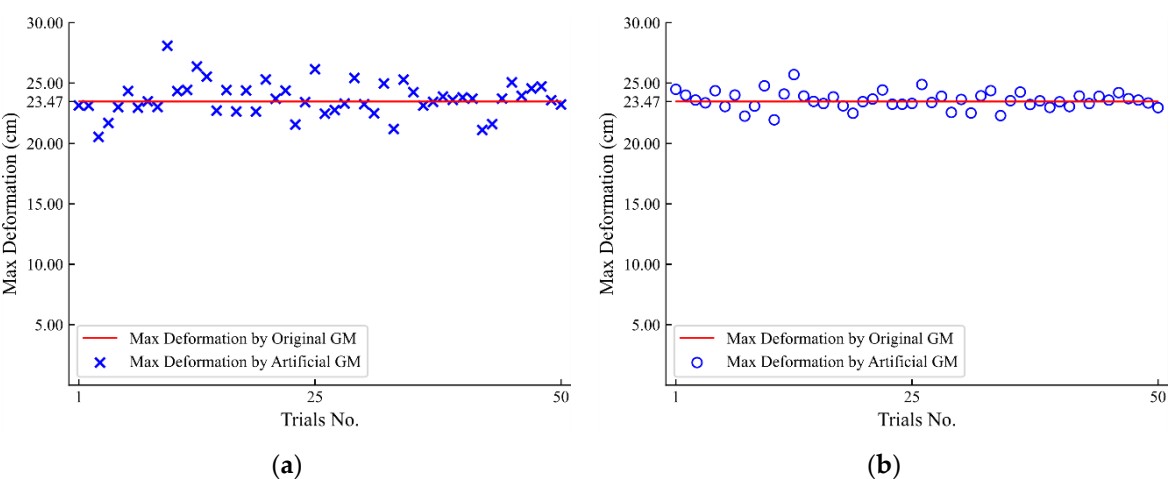

**Figure 10.** Distribution of max response deformations by artificial GMs synthesized using (**a**) CWT and (**b**) DWT.

## 5. Conclusions

For the synthesis of input artificial ground motions for seismic design, there are numerous methodologies based on different theories. In Stochastic Method, Fourier Transform is utilized to disturb the Fourier phase spectrum randomly while the information in frequency domain and the total power of ground motions are maintained. However, localized disturbances at desired time intervals are not available since sinusoidal base function extends in the whole time-domain. By defining complex arguments of the wavelet coefficients as the conception of phase, discrete wavelet transform is introduced to fix such problem, although localized disturbances cannot be arbitrarily oriented due to the property of dyadic distribution of both transition and scale coefficients.

In the article, a new methodology based on continuous complex wavelet transform is proposed, by which localization of phase disturbance in time-frequency domain with arbitrary orientation is realized. Moreover, the computational performance is required at a lower rate and more candidate wavelets, which is not analytic, bases become available in the scheme, enabled by the modified algorithm of inverse wavelet transform. Results of numerical simulation indicate that by utilizing the methodology, random phase disturbance could be precisely localized in the desired time-frequency domain, while variability and diversity of input ground motions are considerably enhanced. For the selection of wavelet candidates, complex Shannon wavelet is evaluated as a better choice for the methodology since the total power and frequency characteristics are highly maintained with random phase perturbation.

For future research, more case studies are supposed to be conducted in order to evaluate the proposed scheme explicitly referring to engineering practice.

**Author Contributions:** Conceptualization, R.H. and H.X.; methodology, H.X. and R.H.; validation, H.X.; formal analysis, H.X.; investigation, H.X.; data curation, H.X.; writing—original draft preparation, H.X.; writing—review and editing, H.X.; visualization, H.X.; supervision, R.H. All authors have read and agreed to the published version of the manuscript.

**Funding:** This research was funded by "Major Program of National Natural Science Foundation of China, grant number 12032008" and "National Key Research and Development Program in China, grant number 2017YFC0806009".

**Data Availability Statement:** Some or all data, models, or code generated or used during the study are available from the corresponding author by request.

**Conflicts of Interest:** The authors declare no conflict of interest.

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
