# Peer review of "Arbitrarily Oriented Phase Randomization of Design Ground Motions by Continuous Wavelets"

_infrastructures, doi:10.3390/infrastructures6100144_

Round 1

Reviewer 1 Report

In this paper it is proposed a new methodology based on continuous complex wavelet transform to estimate ground motions to carry out structural dynamic analyses.

The paper is very interesting and current. The methodology, organization and academic level are good. However, some aspects should be improved:

1) Eq. (1): Please, explain all parameters in the text (e.g., fk, tn, etc.). 

2) I do not understand which equations you used to carry out the analyses (e.g., for Figs. 4-5). Please explain it in the text.

3) Line 199: Did you scale, modify or adjust the artificial accelerations (Fig. 6(b))? Please, explain it in the text. For this, I suggest you read the reference a). 

4) Line 210: Provide more details about the nonlinear analysis.

Suggested references:

a) https://doi.org/10.1193/1.3608002  

b) https://doi.org/10.1016/j.compstruc.2014.10.013

c) https://dx.doi.org/10.24200/sci.2018.50699.1824

Reviewer 2 Report

The article presents stochastic simulations of ground movement wavelets used for the analysis of structure dynamics. A methodology was proposed using numerical simulations of a system with two degrees of freedom. a sprint and dynamic analysis was carried out with the use of Shannon and Gabor's wavelet base. It has been shown that the proposed method allows to verify the time-frequency range of disturbances in the Shannon wavelet domain. The article is strictly theoretical and may be difficult for readers who deal with dynamic issues from the practical point of view (even if it is "short communication").
As the main remark, I indicate the need to supplement the presented theoretical issues. Here are some detailed comments, the inclusion of which should improve the quality of the manuscript:
1. Chapter 1: In the introduction, I propose to emphasize the novelty of my own solutions, especially in the discipline of Infrastructure.
2. Chapter 2.2: Formulas (6) and (7) need a better explanation.
3. Chapter 4.1: I believe that Shannon wavelet transform and Gabor wavelet transform should be better explained.
4. Chapter 4.2: How are the mechanical characteristics of the springs? Usually the mean squared error is given as the reliable result for comparisons. Have the authors made such calculations?
5. Chapter 5: What directions are the authors planning to follow on this topic? 
In fact, authors should explicitly refer to engineering practice. 
